# Can Transformers Smell Like Humans?

**Farzaneh Taleb**
Department of Intelligent Systems
KTH Royal Institute of Technology
Stockholm, Sweden
farzantn@kth.se

**Miguel Vasco**
Department of Intelligent Systems
KTH Royal Institute of Technology
Stockholm, Sweden
miguelsv@kth.se

**Antônio H. Ribeiro**
Department of Information Technology
Uppsala University
Uppsala, Sweden
antonio.horta.ribeiro@it.uu.se

**Mårten Björkman**
Department of Intelligent Systems
KTH Royal Institute of Technology
Stockholm, Sweden
celle@kth.se

**Danica Kragic**
Department of Intelligent Systems
KTH Royal Institute of Technology
Stockholm, Sweden
dani@kth.se

## Abstract

The human brain encodes stimuli from the environment into representations that form a sensory perception of the world. Despite recent advances in understanding visual and auditory perception, olfactory perception remains an under-explored topic in the machine learning community due to the lack of large-scale datasets annotated with labels of human olfactory perception. In this work, we ask the question of whether pre-trained transformer models of chemical structures encode representations that are aligned with human olfactory perception, i.e., *can transformers smell like humans*? We demonstrate that representations encoded from transformers pre-trained on general chemical structures are highly aligned with human olfactory perception. We use multiple datasets and different types of perceptual representations to show that the representations encoded by transformer models are able to predict: (i) labels associated with odorants provided by experts; (ii) continuous ratings provided by human participants with respect to pre-defined descriptors; and (iii) similarity ratings between odorants provided by human participants. Finally, we evaluate the extent to which this alignment is associated with physicochemical features of odorants known to be relevant for olfactory decoding.

## 1 Introduction

The human brain receives sensory input from the environment and encodes it into a high-dimensional representation space, forming a perception of the world [1]. Recent studies have significantly improved our understanding of the underlying mechanisms of visual, linguistic, and auditory perception [2–4]. Indeed, there is a significant level of alignment between human response (from neuron to behavior) and activations of deep neural networks when provided with the same stimuli [5–14].

Despite these recent advances, human olfactory perception remains an under-explored topic. There is no single organizing principle that determines the dimensions of odor space, making the characteriza-

38th Conference on Neural Information Processing Systems (NeurIPS 2024).

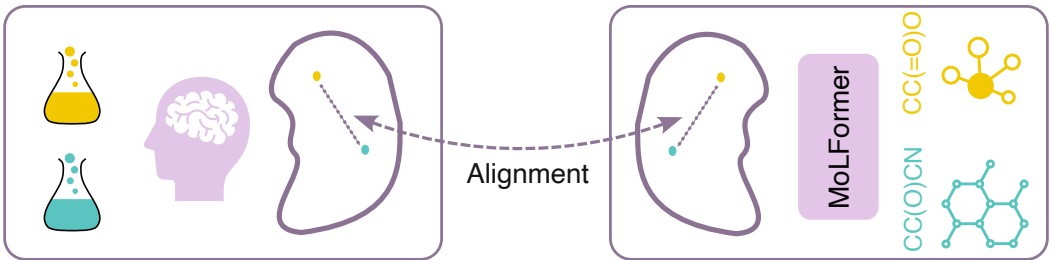

Figure 1: **Evaluating representational alignment** between human and pre-trained transformers. Human participants are stimulated with two odorant substances and asked to rate the perceptual similarity between them (Left). We encode representations of the same pair of odorants using MoLFormer and compute the similarity between pairs of representations (Right). Finally, we measure the alignment between the two systems.

tion of odor perception and its relation to chemical compounds an open and complex problem [15]. A lack of universally accepted methods to describe odorants either quantitatively or qualitatively makes this problem even more challenging. There are very few studies that have explored the mapping of chemical structures to olfactory perception [16–19]. In addition, processing chemical olfactory stimuli using deep neural networks has not been extensively investigated. Nevertheless, training the existing supervised models usually requires an extensive effort by experts to label data.

Transformer-based models [20, 21] are a breakthrough in machine learning, surpassing the need for extensive labeling by utilizing implicit supervision without the necessity for direct labels. These models have demonstrated impressive performance in various tasks such as image [22], video [23], and natural language processing [24]. More recently, they have also shown promising results in encoding chemical structures [25].

In this work, we ask the question of whether representations of odorant chemical structures extracted from transformers pre-trained on chemical structures align with human olfactory perception or, in other words, *can transformers smell like humans*? We employ MoLformer [25], a state-of-the-art transformer, which is pre-trained on chemical structures and we show that representations of odorants extracted from this model:

- can predict *labels assigned to odorants* by experts (Section 4.1);
- can predict *continuous perceptual ratings* provided by human participants (Section 4.2);
- present a high correlation index with *human perceptual similarity ratings* (Section 4.3);
- present a high correlation index with *physicochemical descriptors* known to be relevant for olfactory perception (Section 4.4).

Surprisingly these results hold for models that *were not explicitly trained for the purpose of predicting the human olfactory experience*. To the best of our knowledge, we provide the first empirical study on evaluating the alignment between odorant representations encoded by transformers and human olfactory perception.

## 2   Related Work

The availability of larger datasets, together with advances in predictive methods, has led to an increasing interest in the prediction of olfactory perception from molecular structures.

**Olfactory perception prediction.** Learning predictive models of olfactory from molecular structures has been addressed mostly by the neuroscience community. Several works used standard chemoinformatic representations of molecules to model olfactory perception [17, 19, 18]. Specifically, Snitz et al. [18] proposed a computational framework and algorithm based on structural features of molecules to predict perceptual similarities between odorant pairs. This algorithm leverages feature engineering to identify the most relevant subset of features among 1433 physicochemical descriptors to predict pair-wise odorant perceptual similarities.

Later, Ravia et al. [17] extended this model to also include the perceived intensity of molecular components. They employed 21 physicochemical descriptors discovered in previous works and proposed a weighting approach for multicomponent odorants (MC-odorants) based on their perceived intensity. They reported a higher correlation when employing the weighting approach compared to using the same model without it. However, the representation and generalization capabilities of these models are quite limited and unexplored.

**Deep neural networks for odorants.** Recently, Lee et al. [16] proposed a novel representation learning model of odorants, based on a message-passing graph neural network [26], which they refer to as Principal Odor Map (POM). To train this model, they curated and merged data from `Leffingwell` [27] and `GoodScent` [28] databases to compile a dataset of about 5000 molecules with 138 expert-labeled odor descriptors. This model outperforms the baselines in multiple odor prediction tasks and shows a relatively high alignment with human ratings in describing odorants. Nevertheless, training this model requires labeled data, relying on subjective evaluations of numerous odorants by experts. Besides being time-consuming and laborious, this process can introduce subjective biases into the model, a concern magnified by our incomplete understanding of the foundational factors of olfactory perception.

**Large-scale molecular models.** Large-scale pre-trained models, often known as foundation models [21], have been recently explored to perform diverse tasks by leveraging large amounts of unlabeled data. MoLFormer [25] model has been proposed in the context of chemical prediction tasks, able to extract rich representations from chemical structures. MoLFormer consists of a transformer-based architecture, with linear attention and relative positional encodings. This model is trained using a self-supervised approach, on multiple datasets (e.g., the PubChem [29] and ZINC [30] datasets) on a masked token prediction loss.

## 3 Method

In this section, we provide a detailed description of the datasets utilized in this study and outline the methodology for extracting both odorant (machine) and perceptual (human) representations. Additionally, we present the main model and baseline methods employed, along with the evaluation metrics used to assess their performance. Our experiments do not require significant computational resources: we mostly train linear models that do not involve GPU usage or models that can be trained on a single commercially available GPU under one hour. All computational code to reproduce our results is available at `https://github.com/Farzaneh-Taleb/transformer-olfactory-alignment`

### 3.1 Datasets

We use the publicly available version of the following datasets provided by Pyrfume repository [31].

**Leffingwell-Goodscent (GS-LF) [27, 28].** We employ a curated and merged version of the `Goodscents` [28] and `Leffingwell` [27] datasets, provided by [32], following the procedure introduced by Lee et al. [16]. This dataset contains 4983 molecules with 138 expert-labeled descriptors (e.g. creamy, grassy), where each odorant may be linked to multiple descriptors.

**Sagar [33].** This dataset contains the rating of 160 odorants by 3 human participants, with respect to 15+3 perceptual descriptors. In addition to 15 common descriptors among participants, there are 3 more descriptors that vary among them. We excluded these variable descriptors and focused solely on the common descriptors among the participants. The provided ratings were already normalized within the range of [-1, 1] and the mean ratings across all the subjects are computed for subsequent analysis.

**Keller [34].** This dataset contains ratings of 480 structurally and perceptually diverse molecules by 55 human participants, evaluating 23 descriptors. Participants were instructed to adjust a slider to rate odorants according to individual descriptors, with the slider position subsequently translated into a scale ranging from 0 to 100. Ratings were then averaged across all participants for further analysis.

**Ravia [17].** This dataset contains similarity ratings of 195 unique pairs of MC-odorants and mono-molecules by 94 participants. The similarity values were averaged across all the participants. In this work, we disregarded the factor of odorant intensity and averaged similarity ratings based on the unique pairs of odorants.

**Snitz [18].** This dataset includes similarity ratings from 139 participants and 359 unique pairs of odorants. In each trial, participants were presented with two distinct odorants and asked to rate the degree of similarity in their smells. These ratings were then averaged across all participants.

### 3.2 Odorant representations

Odorants can be described as a single molecule or a mixture of molecules, which we denote as multicomponent odorants (MC-odorants). In this section we describe the method to extract odorant representations from the main pre-trained model (MoLFormer) and our baseline models (DAM and Open-POM).

**MoLFormer.** We employ MoLFormer [25] to encode SMILES strings associated with a single molecule and extract a 768-dimensional vector from the last layer of the model. SMILES (simplified molecular-input line-entry system) is a string-based representation that encodes relevant chemical information such as the type of atoms, their bonds, and the substructures present in the molecule. For MC-odorants, we average the extracted representations across all available mono-molecule components within that MC-odorant. The odorant representation for each dataset is a matrix of $X_i \in \mathbb{R}^{n \times 768}$ where $n$ is the number of unique odorants.

**Open-POM.** The principal odor map (POM) is a supervised-learning model, based on a message-passing graph neural network [26], which is trained on the `GS-LF` datasets to predict odorant perceptual labels. We employ a publicly available version of this model, which we denote by Open-POM [32]. We train Open-POM for 150 epochs, using 30 different train-test splits, and we extract representations from the penultimate layer of this model. The odorant representation for each dataset is a matrix of $X_i \in \mathbb{R}^{n \times 256}$ where $n$ is the number of unique odorants. For MC-odorants, we average the representations extracted for each individual molecule within the mixture.

**Distance Angle Model (DAM).** Snitz et al. [18] proposed a distance angle model (DAM) that uses 21 physicochemical descriptors to predict the similarities between pairs of odorants. We extract these 21 descriptors for each odorant using AlvaDesc [35] and discard 6 of them due to NaN values produced by this software. We use the remaining subset of 15 physicochemical descriptors out of 21 to measure similarity between odorants or train a linear mapping from them to the perceptual representation space. The odorant representation for each dataset is a matrix of $X_i \in \mathbb{R}^{n \times 15}$ where $n$ is the number of unique odorants. As suggested by Ravia et al. [17], we average the representations extracted for each individual molecule within the mixture to compute representations for MC-odorants.

### 3.3 Perceptual representations

Perceptual representations of odorants are provided by human participants when exposed to odorant stimuli. Perceptual olfactory data were collected in one of the following ways:

1. Experts label the odorants, where each odorant may be linked to multiple labels (e.g., [27, 28]). The perceptual representation is a matrix of $y_i \in \{0, 1\}^{n \times d}$ where $n$ is the number of unique odorants and $d$ is the number of classes.

2. Non-expert participants provided ratings with regards to a set of predefined descriptors (e.g., [33, 34].) In this case, the averaged perceptual representations over participants and replicas form a matrix of $y_i \in [a, b]^{n \times d}$, where $n$ is the number of unique odorants, $d$ is the number of descriptors, and $a$ and $b$ denote the minimum and maximum values participants can use to describe the odorants with respect to these descriptors.

3. Participants evaluated the perceived similarity between pairs of odorants (e.g., [18, 17]). In this case, the averaged perceptual representations over participants and replicas are a vector of $y_i \in [a, b]^{n \times 1}$ where $n$ is the number of unique "pair of odorants" and $a$ and $b$ indicate the range of values participants can use to rate the odorants' similarity with respect to the descriptors.

### 3.4 Alignment between perceptual and odorants representations

We measure the similarity between two representation spaces directly when it is possible (Section 4.3), otherwise we train a linear model to predict perceptual representations for each odorant (Section 4.1, 4.2). We use nested 5-fold cross-validation to tune the hyper-parameters of the linear models

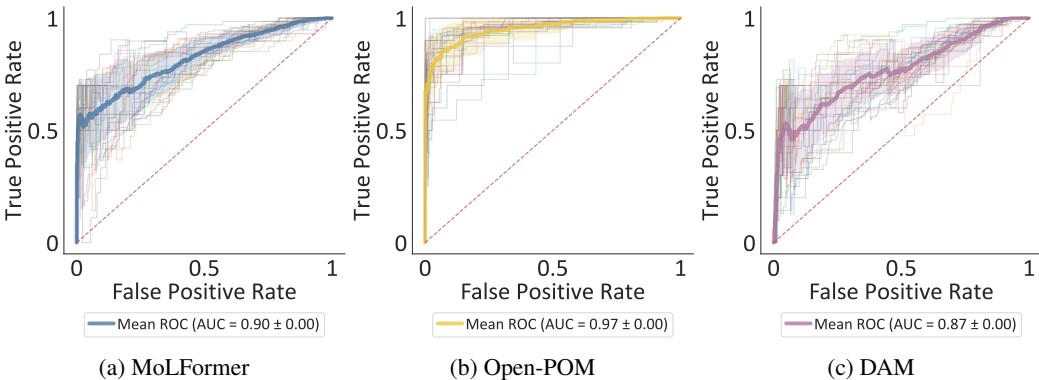

(a) MoLFormer         (b) Open-POM         (c) DAM

Figure 2: **ROC curve for linear classifiers** trained on `GS-LF` representations extracted from three different models. Each curve corresponds to a separate test split, with the thicker curve representing the average performance across all splits. We highlight that MoLFormer outperforms DAM, despite not being trained to predict perceptual labels but does not achieve the performance level of Open-POM, which demonstrates the highest performance. The chance level is shown with red dashed line.

and assess evaluation metrics on the test set that was held out during the training phase using an 80%-20% train-test split. This process is repeated 30 times using 30 different train-test splits.

## 3.5 Evaluation metrics

In this section, we introduce the main evaluation metrics to measure the alignment in this paper.

**Micro-averaged ROC-AUC score.** The micro-averaged ROC-AUC score was computed to assess the performance of each model for the multi-label classification task. The micro-averaged ROC-AUC score is computed by aggregating true positive, false positive, true negative, and false negative values across all classes.

**Normalized Root Mean Squared Error (NRMSE).** The root mean squared error (RMSE) is the difference between the observed values and predicted ones for the regression task. Here, we normalize it by the range of true observations – i.e., $\mathrm{NRMSE} = \mathrm{RMSE}/(\max(y) - \min(y))$.

**Pearson Correlation Coefficient (CC).** We report the Pearson correlation coefficient between predicted results and real values. It measures the linear correlation between two sets of data and is the ratio between the covariance of two variables and the product of their standard deviations.

## 4 Results

In this section, we evaluate whether the representations encoded by pre-trained models of chemical data can predict the human olfactory experience *despite not being explicitly trained for this purpose*. First, we focus on a subset of experiments aimed at predicting expert-assigned labels from odorants through linear mapping from representations to perceptual descriptors (Section 4.1). Subsequently, we aim to predict continuous scores provided by human participants (Section 4.2). Finally, we seek to predict the direct similarity scores from the representations extracted from odorants (Section 4.3). Additionally, we provide insights into the potential reasons underlying the observed alignments (Section 4.4).

## 4.1 Expert-assigned labels classification

To assess the performance of MoLFormer in predicting expert-assigned labels for odorants, we implemented a linear mapping from the representations extracted by MoLFormer to the odorant representations extracted from `GS-LF` dataset. First, the dimensionality of the extracted representations is reduced to 20 using PCA, followed by z-scoring of each feature. Then, we train individual logistic regression models for each descriptor. This process was repeated 30 times, each with a different train-test split, to quantify the uncertainty of the results.

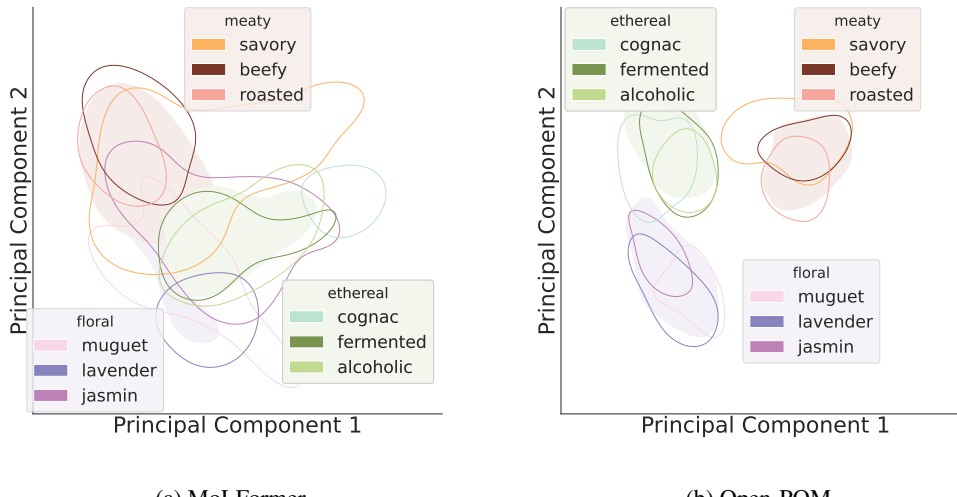

| (a) MoLFormer | (b) Open-POM |

Figure 3: **Visualization of odorant representations** encoded by different models on the `GS-LF` dataset using the figure layout suggested by [16]. We plot the first and second principal components (PCs) of the representation spaces. Areas dense with molecules that have broad category labels (floral, meaty, or ethereal) are shaded, while areas dense with narrow category labels are outlined. MoLFormer captures the perceptual relationship between different odorants in its representation space, despite not being explicitly trained for this purpose.

We apply the same procedure without dimensionality reduction to DAM model representations. For the Open-POM model, which is already trained end-to-end and supervised on the same dataset, we directly extracted the predictions for the test set without retraining the model. As shown in Figure 2 the MoLFormer model achieves high ROC-AUC scores in odorant classification, outperforming the DAM model, which is trained using 15 physicochemical descriptors. However, the performance of MoLFormer is lower than that of Open-POM, which is trained end-to-end with supervision on the same dataset.

An additional experiment is conducted to understand the degree of perceptual details captured in the odorant representation space of MoLFormer by comparing odorant representations encoded by this model with the representations encoded by Open-POM. In Figure 3 we depict the first two principal components of the representations. We highlight the similarity between the representations encoded by both Open-POM and MoLFormer and observe that the latter is able to capture the perceptual relationship between different odorants despite not using any perceptual labels during training (unlike the supervised Open-POM Model).

## 4.2 Continues perceptual rating prediction

To evaluate the capabilities of the MoLFormer model to predict continuous rating scores with respect to pre-defined descriptors, provided by human participants, we train separate linear regression models with regularization applied using the Lasso penalty for each descriptor. Once again, the dimensionality of the extracted representations is reduced to 20 using PCA (for MoLFormer and Open-POM), followed by z-scoring of each feature. This procedure is repeated using 30 different train-test splits.

The results of these experiments are shown in Table 1 and Figure 4. Table 1 shows the average Pearson correlation coefficient and NRMSE across all descriptors, while Figure 4 presents the results for each individual descriptor. As shown in Table 1, overall, none of the models exhibit a high correlation. Nevertheless, MoLFormer slightly underperforms compared to Open-POM in both datasets. However, it performs better than DAM for the Keller dataset but worse than DAM for the Sagar dataset, where DAM even outperforms Open-POM.

Table 1: **Performance of the models to predict continuous ratings averaged across all perceptual descriptors.** We compute the average Pearson correlation coefficient (CC) and normalized root mean squared error (NRMSE) across all descriptors. MoLFormer shows slightly worse performance than Open-POM but better than DAM for the `Keller` dataset and worse than DAM for the `Sagar` dataset, where DAM outperforms Open-POM.

| | Keller | | | Sagar | | |
|---|---|---|---|---|---|---|
| | MoLFormer | Open-POM | DAM | MoLFormer | Open-POM | DAM |
| CC ($\uparrow$) | $0.20 \pm 0.00$ | $0.22 \pm 0.01$ | $0.17 \pm 0.00$ | $0.25 \pm 0.01$ | $0.29 \pm 0.01$ | $0.35 \pm 0.01$ |
| NRMSE ($\downarrow$) | $0.15 \pm 0.00$ | $0.15 \pm 0.00$ | $0.15 \pm 0.00$ | $0.19 \pm 0.00$ | $0.18 \pm 0.00$ | $0.17 \pm 0.00$ |

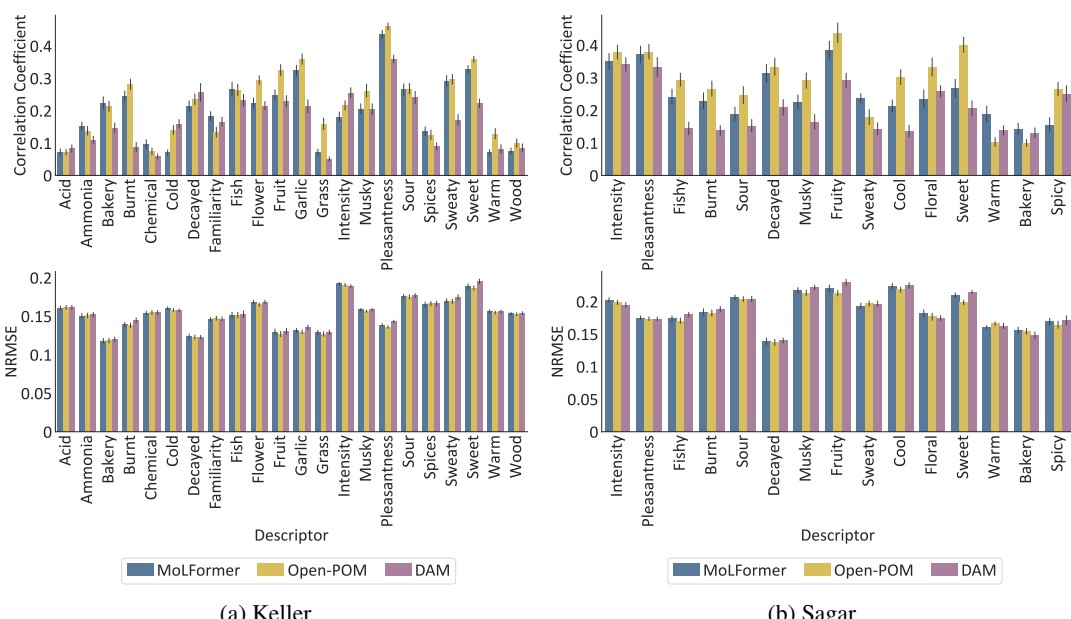

(a) Keller                    (b) Sagar

Figure 4: **Performance of the models to predict continuous ratings per descriptor.** We computed Correlation and NRMSE between predicted and actual ratings per perceptual descriptor. Despite not being trained to predict human olfactory labels, the MoLFormer model performs on par with the Open-POM and DAM models.

According to Figure 4 MoLFormer model performs on par with the Open-POM and DAM models, which are trained with supervision in predicting the rating for each descriptor. In summary, although, on average, MoLFormer performs slightly worse than Open-POM, it still demonstrates a similar degree of alignment, especially despite the absence of supervision in its training process.

### 4.3 Representational similarity analysis

In order to evaluate the direct alignment between the odorant similarities encoded by MoLFormer and those obtained from human participants, we separately encode each odorant by MoLFormer (and the baseline models) and compute the cosine similarity between the extracted representations. Subsequently, we compute the Pearson correlation between the similarity scores computed by the models and those provided by human participants in the `Ravia` and `Snitz` datasets. The results are presented in Figure 5a.

These results show that the MoLFormer is able to extract representations that encode information related to the human olfactory perception, despite not having access to that information during model training. We highlight a significant high correlation between perceptual and odorant representation for the `Snitz` ($r = 0.64, p < 0.0001$) and `Ravia` datasets ($r = 0.66, p < 0.0001$).

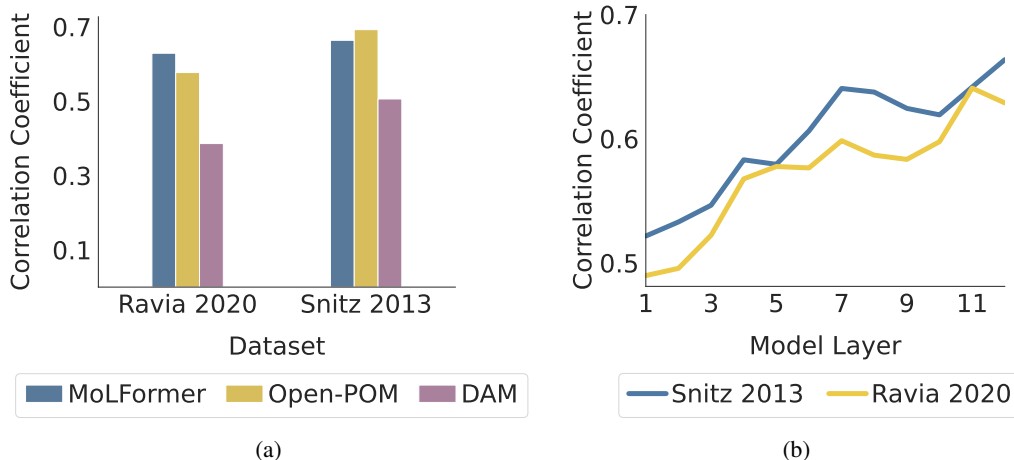

Figure 5: **Representational similarity analysis** for `Snitz` and `Ravia` datasets: a) Correlation coefficients between similarity scores provided by human participants and computed using representations encoded by the different models ; b) Correlation coefficients considering odorant representations extracted from different layers of the MoLFormer model.

The comparison with the baseline models indicates that it performs on par with the Open-POM model and significantly outperforms the DAM model. These results suggest that, despite being trained with some form of supervision, these models may struggle to effectively extract similarities between odorants. Additionally, the findings demonstrate that MoLFormer is more proficient at identifying similarities between pairs of odorants than mapping them to a set of predefined descriptors. This superior performance may be due to the model's ability to capture a measure of similarity, as perceived by humans, rather than introducing subjective language bias associated with pre-defined descriptors.

Finally, we aim to evaluate whether the depth of the layer in the MoLFormer model, from which we extract the odorant representations, affects the representational alignment. To assess this, we repeat the described procedure in this section for each layer separately. As shown in Figure 5b, representational alignment improves with increasing layer depth, indicating that deeper layers of the transformer are more aligned with high-level perceptual representations.

### 4.4 Decoding relevant physicochemical features from pre-trained representations

To evaluate whether MoLformer effectively extracts features from chemical structures relevant to olfactory perception, we evaluate the alignment of MoLFormer with physicochemical descriptors that are used in the DAM model. To do so, we train 15 linear regression models, each one to predict a single physicochemical descriptor from the extracted representations of the MoLFormer. We subsequently evaluate the correlation between the predicted and true values. As shown in Figure 6, MoLformer demonstrates a high degree of alignment in predicting these values. Out of the 15 physiochemical descriptors, MoLformer successfully predicts the values for 13 descriptors as well as or better than the Open-POM model.

Next, we evaluate whether this alignment changes across the layers of MoLformer. Therefore, we repeat the same procedure for each layer separately. As illustrated in Figure 7, the alignment with the identified chemical features decreases with increasing layer depth. However, as demonstrated in Figure 5b, the alignment with perception improves. These results collectively are consistent with well-known principles in vision models, where the lower layers typically capture low-level, localized features like edges and textures, while deeper layers gradually shift toward higher-level, abstract representations, such as shapes and objects[36]. Nonetheless, additional investigation is required to fully reveal and comprehend this potential hierarchical structure.

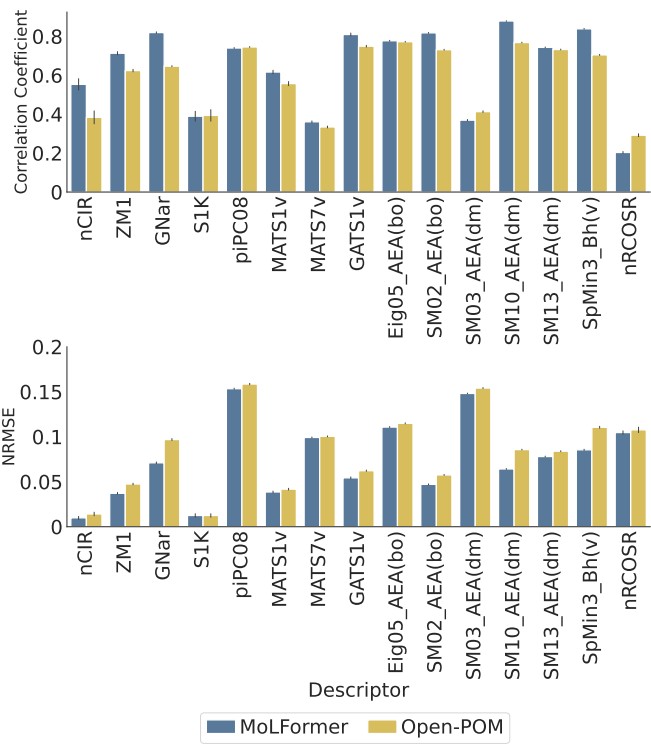

Figure 6: **Performance of the models to predict relevant physicochemical descriptors.** We computed Correlation and NRMSE between the predicted and actual values of descriptors. MoLFormer is able to predict 13 out of 15 physicochemical descriptors related to smell as well as or better than the Open-POM model, demonstrating high alignment with physicochemical descriptors.

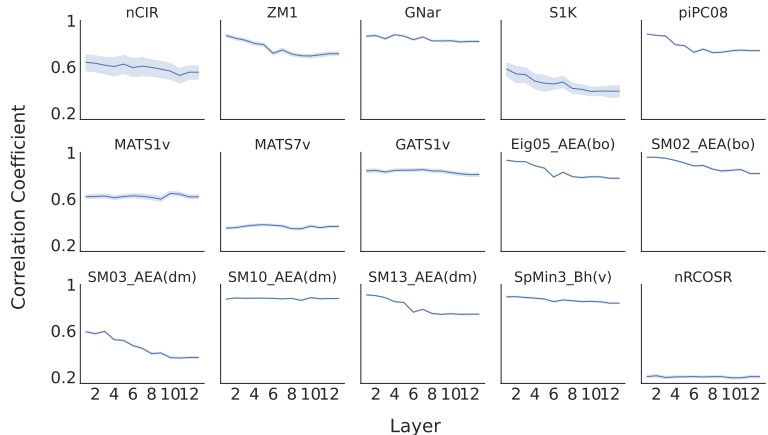

Figure 7: **Correlation between the actual and predicted value of physicochemical descriptors** diminishes as the layer depth increases.

# 5 Discussion

In this study, we investigated the alignment between odorant representations encoded by the MoLFormer, a self-supervised transformer model pre-trained on chemical structures, and human olfactory perception. We evaluated the alignment between these representations by analyzing the similarity between them or finding a linear mapping between the representations. Additionally, we offered insights into the potential reasons behind the observed alignments by exploring relevant chemical features extracted by the model.

**Perceptual prediction from pretrained models**. We demonstrate for the first time that representations extracted from pre-trained large models, solely trained on chemical structures, align closely with the perceptual representations of odorants. This finding suggests that odorant perception can be accurately predicted from chemical structures. Furthermore, we show that this model can predict a subset of physicochemical descriptors known to be relevant to olfactory perception. Together, these results offer valuable predictions for chemists and neuroscientists to explore in future research.

**Evaluating alignment across multiple datasets.** To evaluate alignment from various perspectives, we designed three different experiments. First, we leveraged a dataset with expert-provided labels for odorants, assessing the model's ability to independently predict multi-target binary labels for each odorant. This task did not involve variability from human participants or continuous odorant ratings. MoLFormer exhibited relatively high performance in predicting these binarized labels. Second, we used datasets containing average continuous ratings from human participants, which inherently present more challenges due to variability among non-expert participants' ratings. Our evaluation revealed that while all models performed poorly on this task, MoLFormer performed comparably to supervised models. Lastly, we evaluated direct similarity scores between odorants from two datasets, examining the alignment between human-provided similarity scores and those computed from the representations encoded by models. MoLFormer showed a high alignment, highlighting its ability to predict similarity between odorants rather than relying on human-made descriptors. This suggests that pre-defined descriptors for describing odorants may need to be more carefully chosen, and models trained with these descriptors might not accurately reflect the true similarity between odorants.

**Reduction in alignment with physicochemical descriptors across layers of the models.** We conducted a complementary analysis to identify potential reasons underlying the observed perceptual alignment. Our focus was on the subset of features previously identified as significant for decoding olfactory perception from chemical structures. Our findings indicate that MoLFormer representations exhibit a high degree of alignment with these features. While most features show strong alignment, a few demonstrate less alignment (such as nRCOSR). These results collectively suggest that while these features are important, their significance varies. Additionally, our analysis of the predictability of these features across the different layers of the model shows that as we go through the layers, we observe a decrease in alignment with physicochemical descriptors despite an increase in alignment with perception. This observation aligns well with established principles in vision models, where lower layers have been shown to capture low-level, local features such as edges and textures, while progressively transitioning to align with higher-level, abstract representations, such as shapes and objects, in deeper layers [36]. However, further exploration is needed to fully uncover and understand this potential hierarchy.

**Limitations.** Our work is perhaps best understood in the context of its limitations. We do not directly take into consideration the intensity or concentration of each individual molecule within a mixture during the encoding of odorants. Incorporating these intensity factors in future work could potentially improve the alignment. Additionally, our research was constrained by the available datasets, which typically lack sufficient variations in different odorants, particularly for continuous rating regression tasks. Furthermore, we only considered the average rating scores and did not evaluate the alignment on a per-subject basis.

**Future Work.** We aim to leverage these findings to develop improved models of olfactory perception. Specifically, we plan to utilize unsupervised models trained exclusively on chemical structures to identify which chemical features are crucial for predicting perception, thereby avoiding the introduction of biases from human subjective perception. Additionally, we intend to investigate the mechanisms underlying olfactory perceptions decoded from chemical features. The observed alignment trends across different layers of the model may provide key insights into this process. Finally, evaluating representational alignment between the extracted representations from transformers trained on chemical structures and fMRI data from the brain can provide deeper insights into the underlying mechanisms of olfactory perception.

# Acknowledgement

This work was supported by the Knut and Alice Wallenberg Foundation, Swedish Research Council, ERC AdV BIRD 88480, and ERC Syn D2Smell 101118977. The authors would like to thank Johan Lundström for helpful discussions.

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

# A  Noise Ceiling

In order to evaluate the quality of data and the upper limit of the models, we computed the noise ceiling (Equateion 1, 2) for the `Sagar` and `Keller` datasets as these are the only ones that have multiple evaluators for each odorant, and those are publicly available. The results show that we have the average noise ceilings of $0.28 \pm 0.1$ for the `Keller` dataset and $0.7 \pm 0.05$ for the `Sagar` dataset (Table S.1, S.2). The results show that the data of the `Sagar` dataset is less noisy, and there is still room for the models to increase the alignment. However, the `Keller` dataset alignment results are relatively close to the noise ceiling value.

$$r_j = \text{corr}(\text{Responses from Participant } j, \text{Mean Response across participants}) \tag{1}$$

$$\text{Noise Ceiling} = \frac{1}{N} \sum_{j=1}^{N} r_j \tag{2}$$

Table S.1: Noice ceiling per descriptor for `Sagar` dataset

| Descriptor | Bakery | Burnt | Cool | Decayed | Fishy | Floral | Fruity | Intensity |
|---|---|---|---|---|---|---|---|---|
| **Noise Ceiling** | 0.68 | 0.70 | 0.68 | 0.72 | 0.75 | 0.73 | 0.79 | 0.75 |
| **Descriptor** | Musky | Pleasantness | Sour | Spicy | Sweaty | Sweet | Warm | |
| **Noise Ceiling** | 0.71 | 0.74 | 0.66 | 0.66 | 0.62 | 0.72 | 0.61 | |

Table S.2: Noice ceiling per descriptor for `Keller` dataset

| Descriptor | Acid | Ammonia | Bakery | Burnt | Chemical | Cold | Decayed |
|---|---|---|---|---|---|---|---|
| **Noise Ceiling** | 0.21 | 0.21 | 0.32 | 0.27 | 0.27 | 0.17 | 0.29 |
| **Descriptor** | Familiarity | Fish | Flower | Fruit | Garlic | Grass | Intensity |
| **Noise Ceiling** | 0.33 | 0.21 | 0.26 | 0.37 | 0.31 | 0.25 | 0.53 |
| **Descriptor** | Musky | Pleasantness | Sour | Spices | Sweaty | Sweet | Warm |
| **Noise Ceiling** | 0.22 | 0.52 | 0.23 | 0.24 | 0.24 | 0.41 | 0.17 |

# B  Representational Similarity Matrix (RSM)

In order to better visualize how the models and humans represent different odors, we visualized representational similarity matrices for humans participants, Open-Pom, and MoLFormer across pairs of odorants for `Ravia` dataset in Figure S.1. The white cells show the pair of odorants for which no similarity score is available. MC-odorants corresponding to each mixture are provided in Table S.3

# C  t-SNE Visualizations

We also reduce representations of `GS-LF` dataset extracted from MoLFormer and Open-POM datasets using t-SNE (Figure S.2).

# D  Fine-tuned MolFormer

We fine-tuned MoLFormer using `GS-LF` dataset which is a large and inclusive dataset of odorants. Then we extracted representations for all the datasets and tasks. Figure S.3 shows ROC-AUC curve for `GS-LF` dataset. Figure S.4 demonstrates the results for the continuous rating prediction tasks for `Keller` and `Sagar` datasets, and FigureS.5 shows the results for RSA for `Ravia` and `Snitz` datasets.

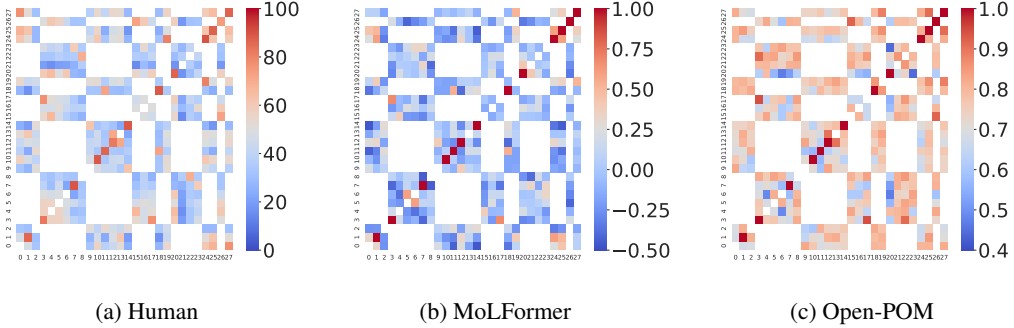

(a) Human        (b) MoLFormer        (c) Open-POM

Figure S.1: RSM for different pairs of odorants for `Ravia` dataset.

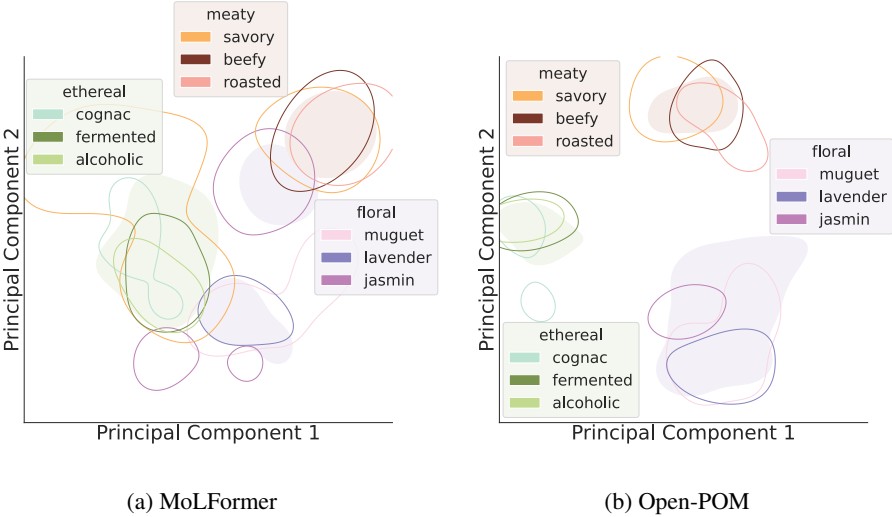

(a) MoLFormer              (b) Open-POM

Figure S.2: **t-SNE Visualization of odorant representations encoded by different models** on the `GS-LF` dataset using the figure layout suggested by [16]. We reduced dimensionality using t-SNE. Areas dense with molecules that have broad category labels (floral, meaty, or ethereal) are shaded, while areas dense with narrow category labels are outlined. MoLFormer captures the perceptual relationship between different odorants in its representation space, despite not being explicitly trained for this purpose.

## E  Decoding chemical features

In this section, we present the results for predicting physicochemical descriptors from odorants for each dataset separately (Figure S.6-S.10). We observe that MoLFormer can better predict physicochemical descriptors in most cases.

## F  Decoding chemical features across layers of MoLFormer

We also show how the alignment between physicochemical descriptors changes across layers of the model for each dataset separately. As it is shown, alignments decrease across layers. This might be because the first layers extract more low-level information while deeper layers, extract more high-level features.

Table S.3: **MC-odorants corresponding of indices in RSM presented in Figure S.1**

| Index | MC-Odorant |
|-------|------------|
| 0 | 126;520296;7122;6050;5273467;5364231 |
| 1 | 1550470;778574;11980;61771;6998;444972;14104;325;23642 |
| 2 | 2214;556940;8180;8077;325;11086 |
| 3 | 240;2758;8130;8129;7710;7059;4133;8918;957;6654 |
| 4 | 240;637511;7731;2758;12178;62336;8635 |
| 5 | 31276;62433;8129;12178;7519;18827;10722 |
| 6 | 31276;8148;7762;18827;7714 |
| 7 | 326;26331;1140;11002 |
| 8 | 5281168;637511;7685;12178;4133;7991;6054;7770;7714 |
| 9 | 5363233;10925;5365049;6050;5273467;31219;7765;23642 |
| 10 | 5363233;89440;126;11980;61293 |
| 11 | 556940;7601;11086;61670 |
| 12 | 565690;8180;5365049;6560;8077;31219;6998;7765;6997;18554 |
| 13 | 62351;1550470;7657;6997;6560;5273467;18554;2214 |
| 14 | 62351;565690;10925;7593 |
| 15 | 62433;8797;2758;3314;8635;61138;11002;6054;10722 |
| 16 | 6544;62433;7519;7685;3314 |
| 17 | 6544;93009;8130;8103;7710;7059;8918;7714 |
| 18 | 7194;520296;61670;637776;23642 |
| 19 | 7194;89440;6560;17121;126;637776;9012 |
| 20 | 7410;240;93009;8635 |
| 21 | 7410;326;2758;62444;7770;1140 |
| 22 | 7410;5281168;8797;7519;8129;7710;6654;8030 |
| 23 | 7519;8148;31252;8103;7710;11002 |
| 24 | 7601;778574;61331;8180;17121;24834;7593 |
| 25 | 7657;61331;61771;61293;24834;31219;444972;5367698;14104;5364231 |
| 26 | 8797;7731;7966;3314;62336;7059;7991;61138;6054;11002 |
| 27 | 89440;7657;7122;61293;7593;5367698;5364231;14104;9012 |

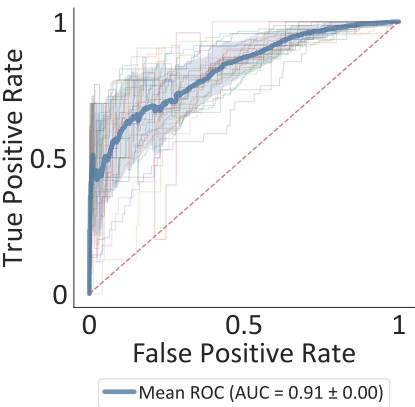

Figure S.3: **ROC curve for the linear classifier** trained on GS-LF representations extracted from the fine-tuned MoLFormer. Each curve corresponds to a separate test split, with the thicker curve representing the average performance across all splits.

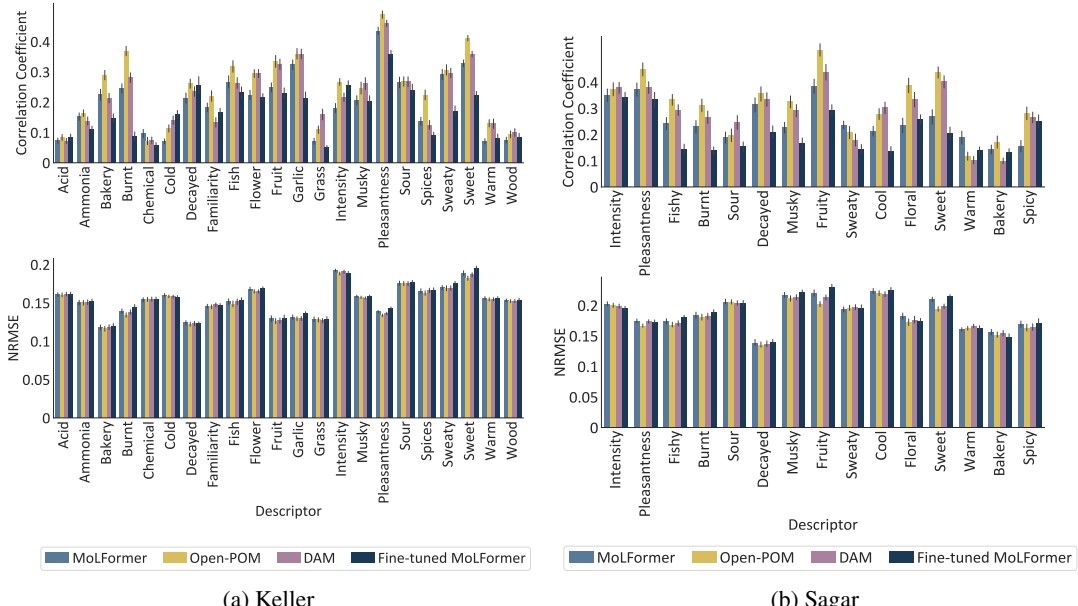

(a) Keller

(b) Sagar

Figure S.4: **Performance of the models to predict continuous ratings per descriptor.** We computed Correlation and NRMSE between predicted and actual ratings per perceptual descriptor. Fine-tuned MoLFormer shows slightly worse performance in predicting continues ratings.

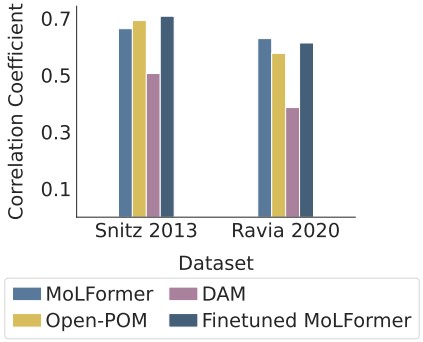

Figure S.5: **Representational similarity analysis** for Snitz and Ravia datasets. Fine-tuned MoL-Former performs on par with MoLFormer.

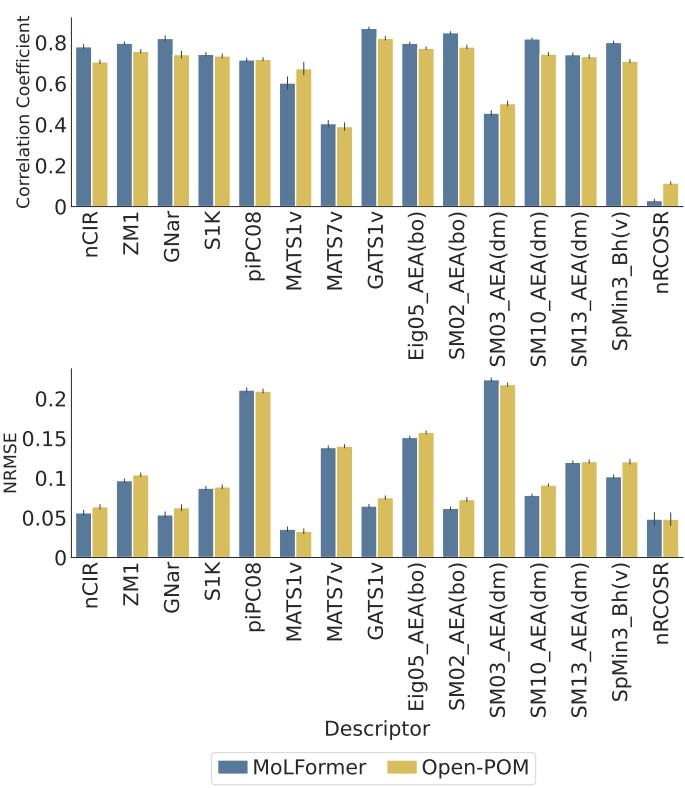

Figure S.6: **Performance of the models to predict relevant physicochemical descriptors** for `Keller` dataset. MolFormer performs slightly better than Open-POM in predicting descriptors.

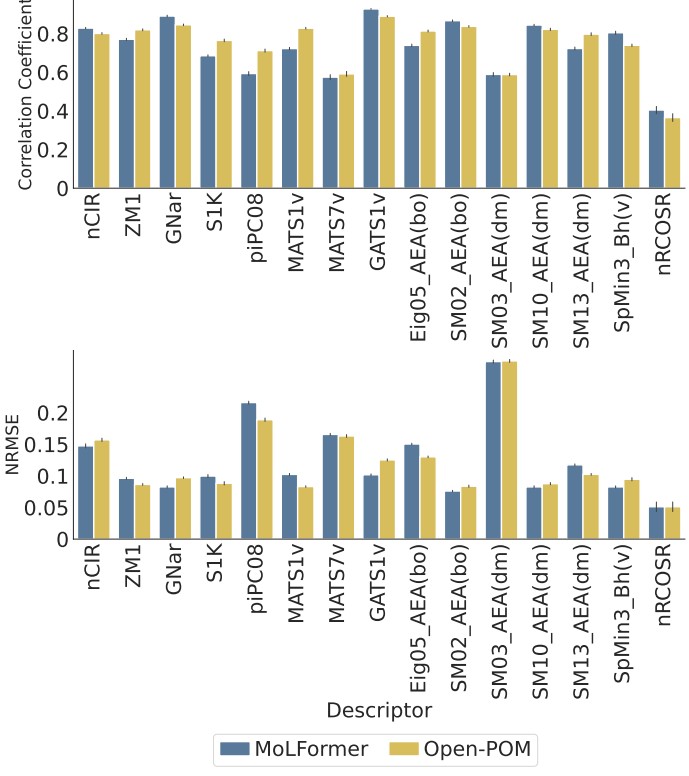

Figure S.7: **Performance of the models to predict relevant physicochemical descriptors** for `Sagar` dataset. MolFormer performs slightly better than Open-POM in predicting descriptors.

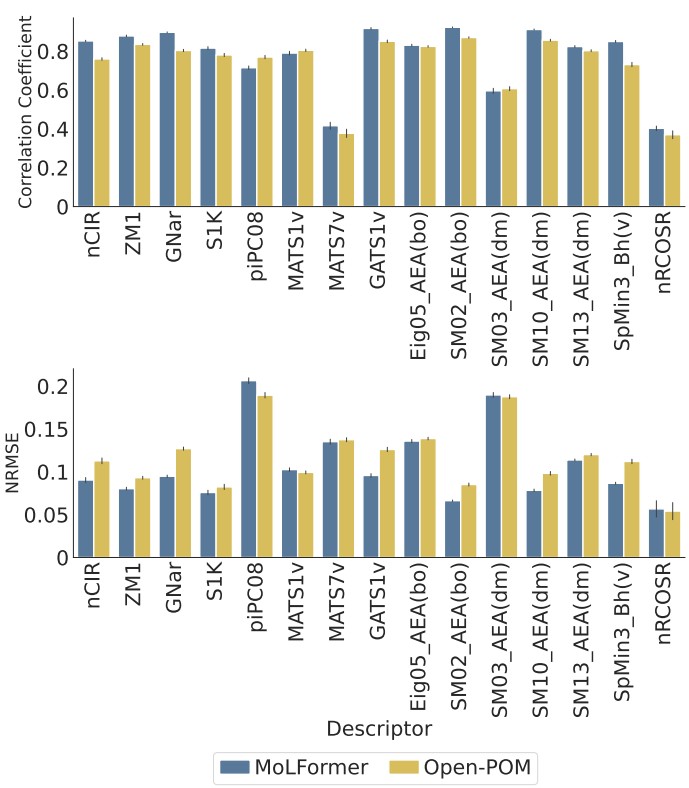

Figure S.8: **Performance of the models to predict relevant physicochemical descriptors** for `Ravia` dataset. MolFormer performs slightly better than Open-POM in predicting descriptors.

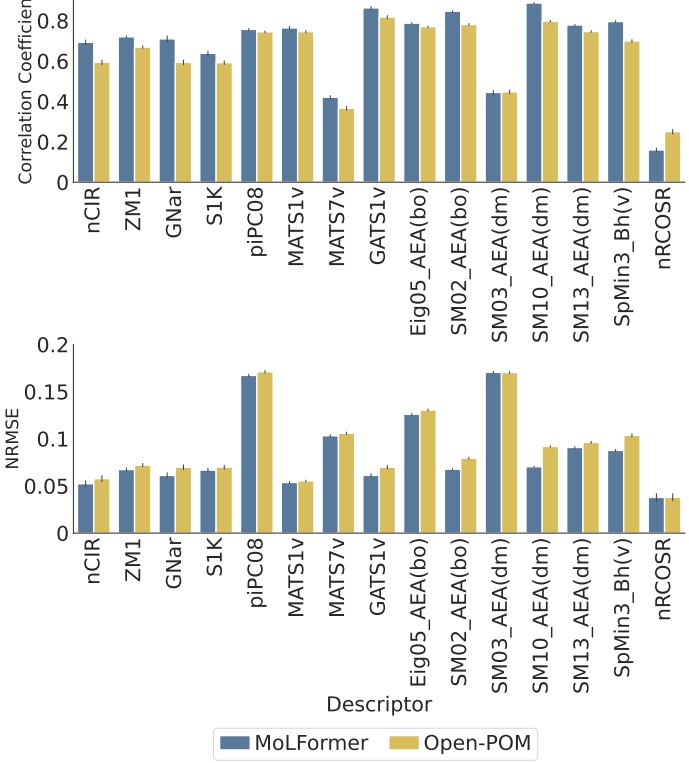

Figure S.9: **Performance of the models to predict relevant physicochemical descriptors** for `Snitz` dataset. MolFormer performs slightly better than Open-POM in predicting descriptors.

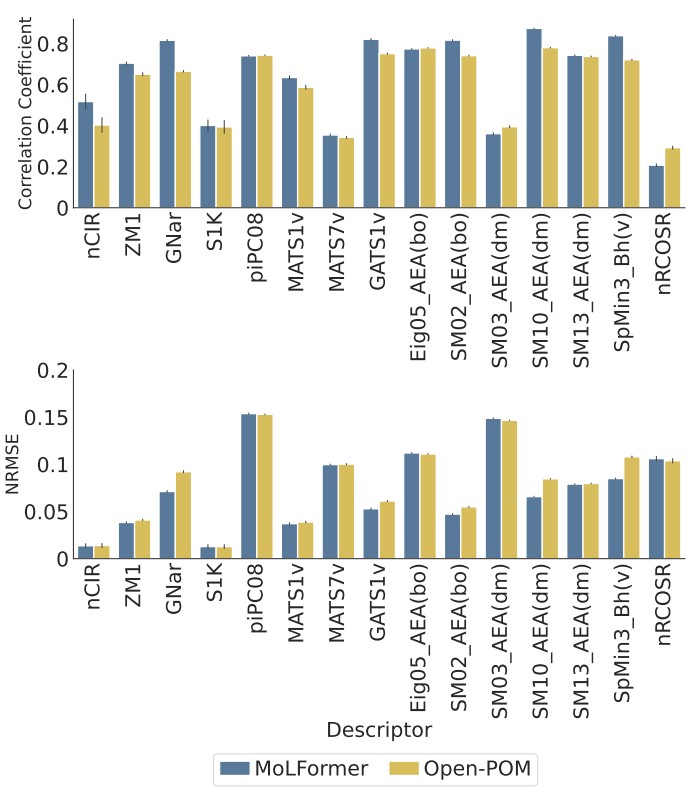

Figure S.10: **Performance of the models to predict relevant physicochemical descriptors** for `GS-LF` dataset. MolFormer performs slightly better than Open-POM in predicting descriptors.

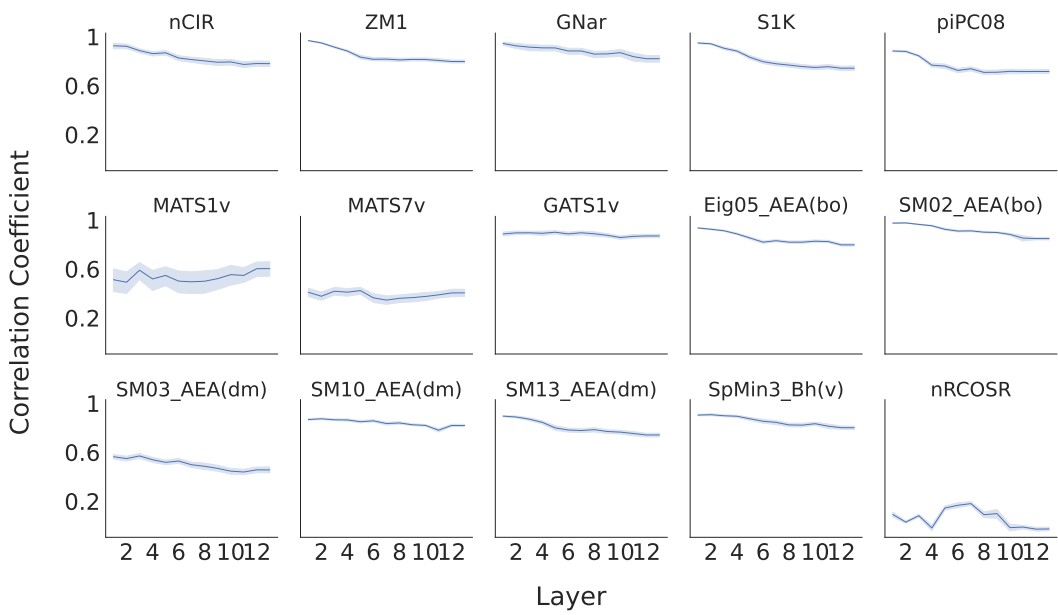

Figure S.11: **Correlation between the actual and predicted value** of physicochemical descriptors in `Keller` dataset diminishes as the layer depth increases .

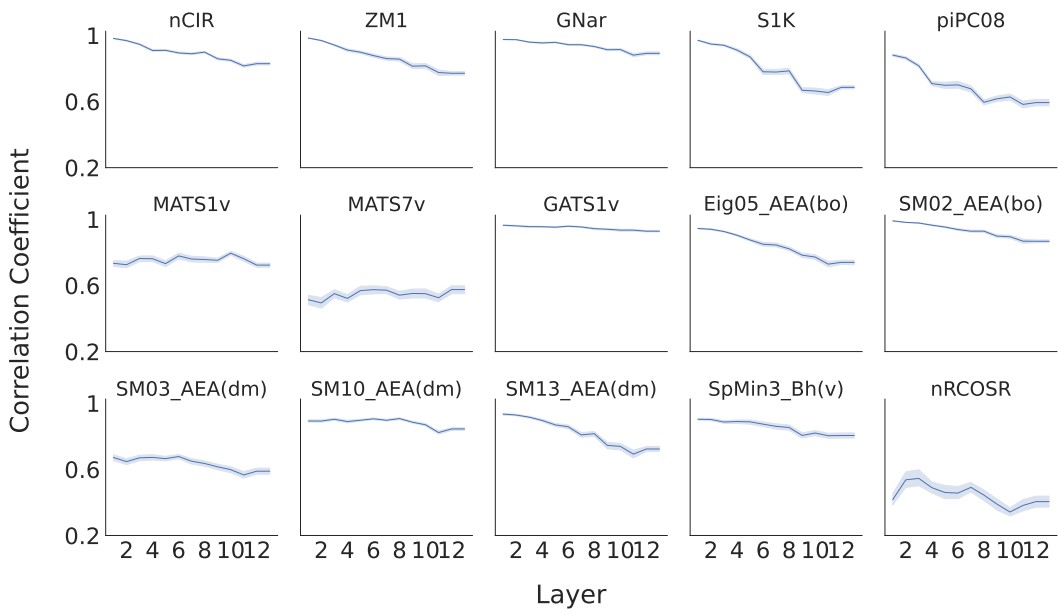

Figure S.12: **Correlation between the actual and predicted value** of physicochemical descriptors in `Sagar` dataset diminishes as the layer depth increases .

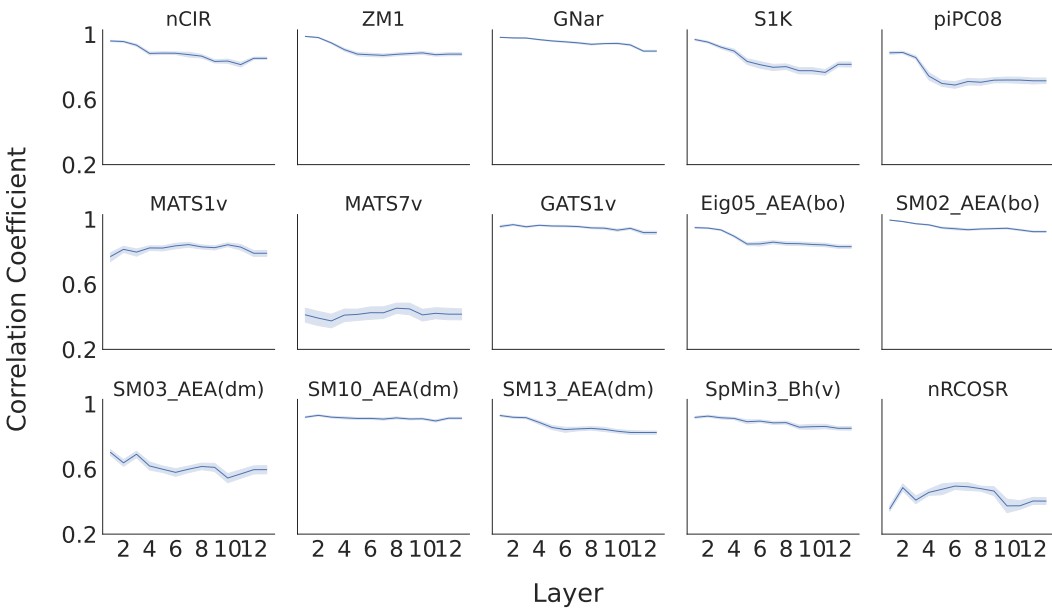

Figure S.13: **Correlation between the actual and predicted value** of physicochemical descriptors in `Ravia` dataset diminishes as the layer depth increases .

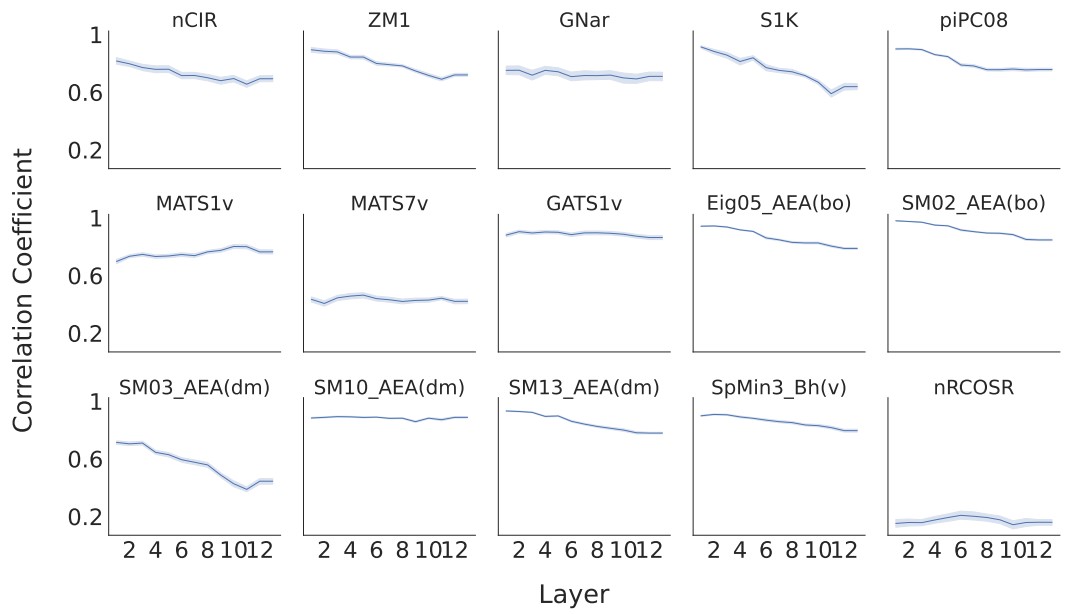

Figure S.14: **Correlation between the actual and predicted value** of physicochemical descriptors in `Snitz` dataset diminishes as the layer depth increases .

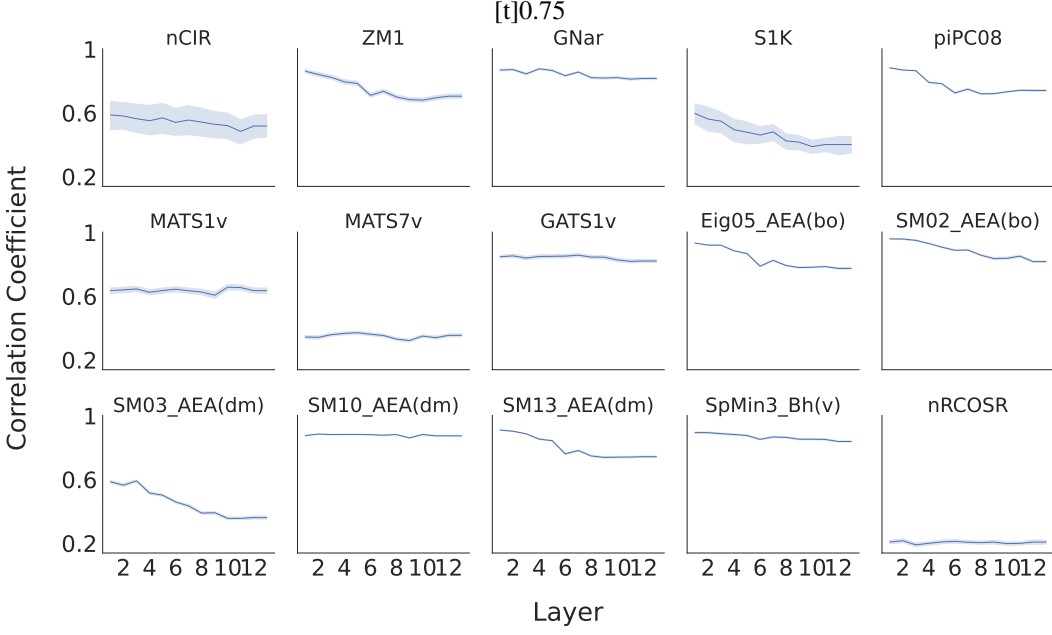

Figure S.15: **Correlation between the actual and predicted value** of physicochemical descriptors in `GS-LF` dataset diminishes as the layer depth increases .

