# OpenReview forum: "Can Transformers Smell Like Humans?"
_NeurIPS.cc/2024/Conference — NeurIPS 2024 spotlight_

### Official Review · Reviewer_6yPX · 2024-07-02

**Soundness:** 3
**Presentation:** 2
**Contribution:** 2
**Rating:** 5
**Confidence:** 4

**Summary:**

This paper mainly focus on the question "Can transformers smell like humans". By using a chemical structure transformer called MoLFormer to encode the odorant molecules, this paper proposes that the MoLFormer pre-trained representation can classify the odors of variety of molecules without careful fine-tuning. The method builds a relationship between odorant chemical strutures and their odors, and the odors are labeled by human-beings.

**Strengths:**

1. This paper tries to use pre-trained transformer to build the relationship between chemical structures and odors, and to some extent, shows that transformer has ability to do odorant classification.
2. This paper applies multiple kinds of databases to support the proposed argumentation, and the databases cover variety of labeling methods to label the odors. Also, experiments are included for every applied labeling methods.

**Weaknesses:**

1. The most important part of the odors classification system is the network, unfortunately, the paper only directly apply a well-designed transformer (MoLFormer) without any modification.
2. It looks like this paper dose not finetune the MoLFormer using the applied databases. The paper argues that the model can achieve good performance without fine-tuning, however, it is better to include experiments to support this argument.

**Questions:**

1. How many models (beside MoLFormer) does the research group consider to generate odorant representations?
2. Do the authors consider to fine-tune the MoLFormer for better performance?
3. Besides transformer, do the authers consider more kinds of models such CNN and graph network?

**Limitations:**

1. It is better to do more analyses about the fine-tuning of the network.
2. It is better to consider to include some innovation in the existing models.

---

> ### Author Rebuttal · Authors · 2024-08-06
>
> We thank the reviewer for the comments and for the time spent reviewing our paper We now address the questions (Q), weaknesses (W), and limitations (L) raised by the reviewer:
>
> * **(W2, Q2,  L1) Fine-tuning results**: We thank the reviewer for the suggestion. We present the results of fine-tuning the complete MoLFormer model in **Figure III** of the general rebuttal document and will add these results to the paper. The change compared to MoLFormer is not significant in many cases. This raises interesting questions regarding the feasibility of fine-tuning pre-trained large-scale chemical models for human olfaction (and the method to do so).
> * **(W1, L2) Model Architecture**:  We agree that the development of novel network architectures for this task is interesting but, as it stands, we believe our work responds to a different, but also very relevant question. We explore whether representations extracted from pre-trained large-scale models on chemical data are aligned with human perception. Our finding suggests that pre-trained models are highly aligned with human perception of odorants **without being explicitly trained** on it, even when compared to supervised learning methods **explicitly trained** on human olfactory perceptual data.
> * **(Q1, Q3) Models employed in our work**: In our work, we consider three types of models to generate odorant representations: (i) the MoLFormer, a self-supervised, *transformer-based*, large-scale pre-trained model of chemical data, which is not trained on human assessments of olfactory stimuli; (ii) the Open-POM, a supervised *graph neural network* model that is trained explicitly using human assessments of olfactory stimuli; (iii) the DAM model, a *feature-engineered* representation model that was specifically proposed to be used in prediction tasks of the human olfactory experience. To the best of our knowledge, Open-POM is the SOTA deep-learning model trained for prediction tasks of the human olfactory experience, and the DAM model is the SOTA feature-engineered model for prediction tasks of the human olfactory experience.

---

> > ### Comment · Reviewer_6yPX · 2024-08-09
> >
> > I thank the authors for answering my comments.
> >
> > I believe that the authors' responses can somehow answer my questions, even though I still cannot be convinced completely. I would like to re-rate the paper to Borderline accept.

---

> > > ### Author Response · Authors · 2024-08-14
> > >
> > > Thank you for the discussion and for recognizing our efforts with the score raise.

---

### Official Review · Reviewer_UQYS · 2024-07-13

**Soundness:** 3
**Presentation:** 3
**Contribution:** 3
**Rating:** 7
**Confidence:** 4

**Summary:**

This paper took a transformer model that was pre-trained on general chemical structures and tested whether the resulting model representations aligned with human olfactory perception. Specifically, the authors used a transformer for chemical structures called MoL-Former. MoL-Former was trained via masked token prediction loss. The representations from MoL-Former can predict odor labels from experts and odor ratings from non-experts (although the correlations between the model and rating scores is quite low). The representations are also highly correlated with physiochemical descriptors that are related to human olfactory perception.

The authors also compare the MoL-Former self-supervised transformer model to other baseline models (Open-POM, which is supervised with odor labels; and DAM, which predicts similarities between pairs of odors). MoL-Former generally performs comparably to these models, even though the baseline models are supervised and MoL-Former is self-supervised.

**Strengths:**

Transformer models have been revolutionary in modeling sensory modalities of vision and audition, but olfaction is comparatively understudied. If olfaction can also be explained with transformer models, it indicates that the domain-general learning mechanisms of transformers can explain sensory processing in the brain more broadly. Put more simply, the results support a universal, domain-general learning mechanism for sensory processing across modalities. This, in and of itself, is an interest finding worthy of publication.

Another strength of the paper is that it relies on self-supervision to pre-train the transformer. This is important both practically and theoretically. From a practical standpoint, acquiring labelled datasets is challenging, whereas acquiring large-scale unlabelled datasets is becoming increasing more feasible. From a theoretical standpoint, this provides a better comparison to human olfaction than supervised models. Most human olfactory learning (and sensory learning more generally) is unsupervised. We are rarely given labels for odors, and animals are able to learn about odors without any verbal labels.

Finally, the inclusion of baseline models for comparison to the unsupervised transformer is important for interpreting their results. By comparing to OpenPOM and DAM, the author demonstrate that a self-supervised transformer can perform comparably to previous supervised models.

**Weaknesses:**

Major
- Section 4.1 Expert labels classification was difficult to follow. How was dimensionality reduced to 20, through PCA perhaps? How was the same procedure applied to DAM, given that DAM already has less than 20 features? Why wasn't the visualization in Figure 3 also performed on DAM?
- Sections 4.2 & 4.3 need noise ceilings in order to be interpretable. The authors analyzed each model's predicted average rating with the actual average rating from participants. However, it is unreasonable to expect a model to be more correlated with average ratings than a typical individual human participant. The noise ceiling computes the average correlation between the human participants and mean performance to estimate a reasonable "ceiling" performance that could be expected from models. When the data are very noisy, the noise ceiling will be low (and the human benchmark will generally be less useful because of its low precision), and when the data are not noisy, the noise ceiling will approach 1.

Minor
- The first sentence of the paper references Damasio, 1989 and Meyer & Damasio, 2009. These are both fantastic papers, but they don't really fit the sentence where they are referenced. References like DiCarlo & Cox, 2007 and Olshausen & Fields, 2004 strike me as more relevant.
- A representational similarity matrix (RSM) would be a great visualization for Section 4.3. It would be really interesting to visualize how the patterns of similarity between different odorants compared across models and humans. You don't need to add it, but it would be helpful for readers to visualize how the models and humans are representing the different odors.
- The description of SMILES is a bit hard to follow. Could you provide an example and explain how (or if) the inputs to Open-POM, DAM, and MoL-Former differ?

**Questions:**

Does the input to MoL-Former (and the baseline comparison models) correspond to olfactory receptors? If the input data from olfactory receptors does not match the input data to MoL-Former, then it's not actually completing the same task, and it becomes much harder to argue that the model can "smell like humans."

**Limitations:**

See weaknesses above.

---

> ### Author Rebuttal · Authors · 2024-08-06
>
> We thank the reviewer for the comments and for the time spent reviewing our paper. We now address the questions (Q) and weaknesses (W) raised by the reviewer:
> * **(W1.1) Dimensionality Reduction**: In this work, we use PCA to reduce the dimensionality of the representations for MoLFormer and Open-POM. You are correct that DAM does not require PCA. We have clarified this in the updated version of the document.
> * **(W1.2) DAM on Figure 3**: We thank the reviewer for the comment. We have added the t-SNE visualization of the DAM model **(Figure II. c)** in the general rebuttal document. We also added such visualizations using UMAP and PCA in the updated version of the paper. (We didn’t include them in the general rebuttal document due to the lack of space.)
> * **(W2) Noise ceilings**: We thank the reviewer for the suggestion to use noise ceilings in Section 4.2 and Section 4.3. We have computed the noise ceilings for the "Sagar" and "Keller" datasets (as these are the only ones that have multiple evaluators for each odorant, and those are publicly available.), as shown in the following tables. The results show that we have noise ceilings of **$0.28\pm{0.1}$** for the Keller dataset and of **$0.7\pm{0.05}$** for the Sagar dataset. The results show that the data of the Sagar dataset is less noisy, and there is still room for the models to increase the alignment. However, the Keller dataset alignment results are relatively close to the noise ceiling value. We will also add the results divided by noise ceiling value in the appendix of the paper.
>
> **Sagar dataset**:
>
> | Bakery | Burnt  | Cool | Decayed | Fishy|Floral | Fruity | Intensity | Musky | Pleasantness | Sour| Spicy | Sweaty | Sweet | Warm |
> |:-------------:|:-------------:|-------------:|-------------:|-------------:|-------------:|-------------:|-------------:|-------------:|-------------:|-------------:|-------------:|-------------:|-------------:|-------------:|
> | 0.68  | 0.70 | 0.68| 0.72| 0.75 |0.73 | 0.79| 0.75| 0.71   |  0.74| 0.66|  0.66  | 0.62|0.72| 0.61|
>
>
>
>
>
>
>
>
>
>
>
> **Keller dataset**:
> |Acid|Ammonia|Bakery|Burnt|Chemical|Cold|Decayed|Familiarity|Fish|Flower|Fruit|Garlic|Grass|Intensity|Musky|Pleasantness|Sour|Spices|Sweaty|Sweet|Warm|
> |:-------------:|:-------------:|:-------------:|:-------------:|:-------------:|:-------------:|:-------------:|:-------------:|:-------------:|:-------------:|:-------------:|:-------------:|:-------------:|:-------------:|:-------------:|:-------------:|:-------------:|:-------------:|:-------------:|:-------------:|:-------------:|
> |0.21|0.21|0.32|0.27|0.27|0.17|0.29|0.33|0.21|0.26|0.37|0.31|0.25|0.53|0.22|0.52|0.23|0.24|0.24|0.41|0.17|
>
>
>
> * **(W3) Introduction references**: We do agree that DiCarlo & Cox, 2007 and Olshausen & Fields, 2004, are more suitable for the purpose of this sentence and, as such, we have followed the reviewer’s suggestion and replaced them in the updated version of the paper.
>
> * **(W4) RSM analysis**: We thank the reviewer for the analysis suggestions. We provided the RSM figure only for the Ravia dataset and for MoLFormer, OpenPom and Human Perception in **Figure I** of the general rebuttal document, and we will add that for both the Ravia and Snitz datasets and also for the DAM model in the updated version of the paper (we don't include Snitz dataset and DAM model in the pdf due to the lack of space). We also added a table showing the correspondence of labels on axes with the real CIDs in the paper appendix.
> * **(W5) SMILES/Input for the different models**: The SMILES representation encodes the three-dimensional structure of a chemical (e.g., a molecule) into a short ASCII string. It does so by first performing a depth-first tree search over the chemical graph, adding the elements, bonds, and rings, and subsequently removing hydrogen atoms and breaking cycles (e.g., water is represented by “O” in the SMILES notation). We use SMILES representations of odorants as input to the MoLFormer. For example, toluene is written as a string like "Cc1ccccc1". For the Open-POM, each odorant is represented as a graph, where each atom is represented as a node feature vector (e.g., valence, hydrogen count, atomic number…), and each bond is represented as a edge feature vector (e.g., degree, aromaticity, whether it is a ring or not). Finally, for the DAM model, we use a set of predefined physio-chemical features, which we extract using the AlvaDesc software. We have clarified these representations in the updated version of the document.
>
> * **(Q1) Correspondences of Inputs**: We thank the reviewer for raising this point. In our work, “smell like a human” corresponds to whether representations of odorant chemical structures extracted from learning-based models are aligned with human olfactory perception (line 37). The underlying assumption to this question is that the "stimuli" provided to human participants are similar to the ones provided to the models using different types of representations. We agree that there are limitations that are not captured by our representation, such as the concentration and intensity of different molecules in an odorant, and we will try to discuss this better in the updated paper. But in our study, input to the olfactory receptors are the molecules of a specific odorant, and we used the representations of the exact same molecules as the input of the model.

---

> > ### Comment · Reviewer_UQYS · 2024-08-12
> >
> > Thank you for providing these helpful explanations and revisions!
> >
> > Any idea why there isn't much visible grouping by scent category (e.g., meaty, floral, ethereal) for the RSM visualization? I suppose the grouping from PCA, t-SNE, and UMAP must rely on how different features are weighted and combined.
> >
> > Regardless of that question, I am very enthusiastic that the Sagar noise ceiling is reasonably high. You mention that you will "add the results divided by noise ceiling value in the appendix of the paper," and I ask that you please also include what the noise ceilings are for reference. This gives readers important information about the quality of each benchmark dataset.
> >
> > Based on the authors' rebuttal PDF and responses to my review, I have decided to raise my score. I will modify my review to raise my rating from a 6 to a 7.

---

> > > ### Author Response · Authors · 2024-08-14
> > >
> > > We will add the definition of noise ceilings in the Appendix. Nice suggestion!
> > >
> > > Interesting question about the RSM visualizations. Your hypothesis is an interest one. Indeed, we believe that the main directions used in PCA and the different ways the features are combined in t-SNE and UMAP are quite important in making these groups clearly distinct.
> > >
> > > Finally, we thank the reviewer for the discussion and feedback. Also, thank you for recognizing our efforts with the score raise.

---

### Official Review · Reviewer_Y4G5 · 2024-07-13

**Soundness:** 4
**Presentation:** 2
**Contribution:** 3
**Rating:** 7
**Confidence:** 5

**Summary:**

This paper investigates the ability of MOLformer, a model trained on SMILES strings in a BERT-like fashion, to predict the human assessment of odors. The assessments were based on natural language descriptors of odors by human experts, ratings on different NL descriptors by naive subjects, and finally, on similarity ratings between odors.  It compares MOLformer, which was trained in an unsupervised manner, to an engineered model (DAM) and a supervised model Open-POM. Despite being trained in an unsupervised manner to represent molecules, MOLformer holds its own against the supervised models. In addition, MOLFormer shines when evaluated using an RSA technique, suggesting that MOLformer captures the relationships between molecular odors better than the other models.

**Strengths:**

+ Assessing models representations of odors is an understudied area. I believe this is the first work to do this.
+ quite a few datasets were used, providing binary, rating scale, and similarity judgment data. These might be the only ones available
+ The results are relatively convincing

**Weaknesses:**

+ Despite a rather systematic reporting of models and datasets, some things were left out of the descriptions. For example:
	- Apparently the GS-LF datasets features are binary. I assume this means that that particular descriptor was used by at least 1 judge. So is it the case that if multiple judges used the same descriptor, the results were still recorded as binary?
	- how dimensionality was reduced. I assume PCA. How much variance was captured in 20 PCs?
	- There were apparently descriptors in the similarity dataset as well (line 123). These weren’t described.

+ the references in the introduction were weird. See below.

Typos, comments:

The references you use in the introduction seem rather bizarre. Two Damasio references for encoding sensory input into a high-dimensional space?
Reference #4 seems out of place in this context.
References 8-10 - you have no primary references here; for example, most would reference DiCarlo’s lab here.
Reference #12 is weird. Why not reference the original transformer model? And while it’s true that for language, they are self-supervised, the original use of them for vision was supervised. It is only later that self-supervised vision models (e.g., the Masked Autoencoder) came about.

line 53: Olfactory is mis-spelled!

Figure 2: You might try using UMAP or t-SNE here instead of (or in addition to) PCA.

In your results section, you never say what the train-test split was (i.e., 80/20?)

I was thinking - they should use RSA - and then you did! This is a very interesting result because, rather than just being able to predict human results, the model is better at both of the others in capturing the *relationships* between the odors, just from their chemical descriptions.

- Figure 6: Explain why didn’t you evaluate DAM here.
- You didn’t describe how the physicochemical descriptors were obtained.

Line 284: In section 4.4, it wasn’t clear that for this analysis, you were trying to discover the reasons for the alignment. You should say something about that when introducing that analysis.

lines 289-292: It makes sense to me that the low-level features of the molecules would be best encoded in earlier layers and not in later layers, where more abstract features would be encoded. This seems to fit well with the way vision models work.

**Questions:**

See my questions above:
1. variance captured by PCA
2. How are the physicochemical descriptors derived?
3. Try UMAP or t-SNE

**Limitations:**

These are discussed.

---

> ### Author Rebuttal · Authors · 2024-08-06
>
> We thank the reviewer for the comments and for the time spent reviewing our paper. We address the questions (Q), weaknesses (W), and comments (C) raised by the reviewer as follows:
>
>
> * **(W1) Description of GS-LF dataset**: GS-LF dataset is a multi-label binary dataset, where each data point (the “odorant”) has a set of labels (the “descriptors”) evaluated by a single expert evaluator (the “judge”). As such, no odorant has ratings from multiple judges, but the single judge was asked to assign labels to each odorant chosen among 138 descriptors. We clarified this point in the updated version of the paper.
>
> * **(W2 and Q1) Dimensionality reduction method and variance captured**:  In this work, we use PCA to reduce the dimensionality of the representations. We report the amount of variance explained by the first 20 PCs in the following table. The results show that our 20 PCs capture the majority of the variance of the data.
>
> | Dataset / Method | Keller | Sagar | GS-LF|
> | -------------|:-------------:|:-------------:|:-------------:|
> |MoLFormer|${0.62 \pm{ 0.002}}$ |${0.67\pm{0.004}}$|${0.59\pm{0.000}}$|
> |Open-Pom|${0.70 \pm{ 0.002}}$|${0.74\pm{0.003}}$|- (trained end-to-end)|
> |Fine-tuned MoLFormer|${0.75\pm{ 0.001}}$|${0.78 \pm{ 0.003}}$|${0.75\pm{0.03}}$|
>
>
>
> * **(W3) Descriptors in the similarity dataset (line 123)**: You are correct. Line 123 is not accurate. Similarity datasets do not include descriptors, and in this case, $y_i \in [a, b]^{n  \times 1}$. We have corrected this line in the updated version of the paper.
> * **(W4) References in the introduction**: We thank the reviewer for this comment. As pointed out as well by reviewer UQYS, we have replaced Damasio references by the work of DiCarlo & Cox, (2007) and Olshausen & Fields, (2004). We agree that reference [4] is out of place: in line 34 it should read “(...) impressive performance in various tasks such as control [4], image [13], (...)”. We have also added the reference to the work of Vaswani et al. (2017) as the original reference for the transformer. Moreover, we agree that [13] uses a supervised objective to train the transformer, and we also added the reference to He et al. (2022) on MAEs to the discussion on self-supervised image transformers.
> * **(C1, Q3) UMAP and T-SNE visualizations**: We add the T-SNE figures to the attached rebuttal PDF. We also ran experiments with UMAP and will add it to the appendix of the paper (we don't include in the attached PDF due to lack of space). Overall, we found these figures to be harder to interpret than the PCA figure, which was already suggested in [20].
> * **(C2) Train-test split**: In all evaluations in our work, we use a 80-20% train-test split, with 30 different randomly-seeded partitions. Moreover, we used nested cross-validation to estimate the hyper-parameters of the linear models. We have added this information to the beginning of Section 4 of the updated version of the paper.
> * **(C3, Q2) Physio-chemical descriptors and Figure 6**: We extract the physio-chemical descriptors using the AlvaDesc software. These physio-chemical descriptors are those that are used as input features of the DAM model proposed by Snitz et al. (2013). As such, in Figure 6, we don’t evaluate the prediction capabilities of the DAM model, as the targets would be the same as its input features. We have clarified this point in Section 4.4 of the updated version of the paper.
> * **(C4) Motivation of section 4.4 (line 284)**: We have clarified the motivation for the additional analysis in the updated version of the paper.
> * **(C5) Alignment with low-level physio-chemical descriptors**: We thank the reviewer for the insightful comment on the results regarding the alignment of the model with physio-chemical descriptions, which we fully agree with. The reviewer pointed out correctly that in vision models, low-level features are extracted in the early layers of the model, and the same behavior can be observed here. We added this insightful interpretation in our final paper.

---

> > ### Comment · Reviewer_Y4G5 · 2024-08-11
> > **thanks for the response**
> >
> > Thanks for your response to my criticisms. Since I already gave the paper what I consider a relatively high score, I will leave it at that. I think it's a very interesting paper, showing that a self-supervised transformer trained on molecular structures (without anything specific to odors), can reasonably well predict odor ratings by humans. Good job, and I hope the paper is accepted.

---

> > > ### Author Response · Authors · 2024-08-14
> > >
> > > Thank you for your comments and feedback

---

### Author Rebuttal · Authors · 2024-08-06

We thank all the reviewers for the constructive and interesting questions and also for the positive and encouraging feedback. We did the following extra experiments, analysis, and visualizations to answer the questions raised by the reviewers.

1. We visualized representational similarity matrices (RSM) for Ravia and Snitz datasets. **Figure I** shows results for the Ravia dataset
2. We applied two additional methods for reducing dimensionality: t-SNE and UMAP. **Figure II** shows the results obtained for t-SNE.
3. We fine-tuned the model and visualized the results of fine-tuning in the general rebuttal document, see **Figure III**.

You can find these results in the attached document.

---

### Decision · Program_Chairs · 2024-09-25

**Decision:**

Accept (spotlight)

**Comment:**

The work described in this paper demonstrates that a transformer model that was pre-trained on general chemical structures (self-supervised) forms representations that align well with human olfactory perception.

Olfaction is still understudy (compared to e.g., vision or audition) and our understanding of the neural computational processes and
representations underlying human olfactiory perception is still very limited. Thus this paper makes a welcomed contribution. The reviewers
liked the model approach, the careful comparison with mutliple datasets of human psychophysical assessments of odors, and the results. The work suggests that the domain-general learning mechanisms of transformers may explain sensory processing in the brain more broadly, which is an interesting scientific hypothesis with broad implications. Thus, the work is of substantial interest to the NeurIPS community.